# Physicochemical Properties and Antioxidant Activity of Pumpkin Polysaccharide (*Cucurbita moschata* Duchesne ex Poiret) Modified by Subcritical Water

**DOI:** 10.3390/foods10010197

**Published:** 2021-01-19

**Authors:** Guoyong Yu, Jing Zhao, Yunlu Wei, Linlin Huang, Fei Li, Yu Zhang, Quanhong Li

**Affiliations:** 1Beijing Advanced Innovation Center for Food Nutrition and Human Health, College of Food Science and Nutritional Engineering, China Agricultural University, Beijing 100083, China; yuguoyong66@163.com (G.Y.); zhaojing_cau@hotmail.com (J.Z.); wei_yunlu@163.com (Y.W.); linlinhuang0605@163.com (L.H.); pdlifei@163.com (F.L.); hnzzzhangyu@163.com (Y.Z.); 2National Engineering Research Center for Fruit and Vegetable Processing, Beijing 100083, China; 3Key Laboratory of Fruit and Vegetable Processing, Ministry of Agriculture, Beijing 100083, China; 4Beijing Key Laboratory for Food Non-Thermal Processing, Beijing 100083, China

**Keywords:** subcritical water, pumpkin polysaccharides, physicochemical characteristics, antioxidant

## Abstract

In this paper, subcritical water (SCW) was applied to modify pumpkin (*Cucurbita moschata* Duchesne ex Poiret) polysaccharides, and the properties and antioxidant activity of pumpkin polysaccharides were investigated. SCW treatments at varying temperature led to changes in the rheological and emulsifying properties of pumpkin polysaccharides. SCW treatments efficiently degraded pumpkin polysaccharides and changed the molecular weight distribution. Decreases in intrinsic viscosity, viscosity-average molecular weight, and apparent viscosity were also observed, while the activation energy and flow behavior indices increased. The temperature of SCW treatment has a great influence on the linear viscoelastic properties and antioxidant activity of pumpkin polysaccharides. Pumpkin polysaccharides solution treated by SCW at 150 °C exhibited the highest emulsifying activity and antioxidant activity, which was probably due to a broader molecular mass distribution and more reducing ends exposed after treatment. Scanning electron microscopy showed that SCW treatment changed the microstructure of pumpkin polysaccharides, resulting in the exposure of bigger surface area. Our results suggest that SCW treatment is an effective approach to modify pumpkin polysaccharides to achieve improved solution properties and antioxidant activity.

## 1. Introduction

Pumpkin (*Cucurbita moschata* Duchesne ex Poiret) is a popular cultivated vegetable that exhibits a range of promising health-promoting properties [1,2]. Pumpkin polysaccharides are one of the primary functional and nutritional components derived from pumpkins [3], and they exhibit a range of antitumor, antioxidant, antibacterial, antidiabetic, and antiobesity properties [2,3,4,5,6,7,8,9,10]. Polysaccharides have also been leveraged as stabilizing, thickening, binding, and emulsifying agents in the industrial production of foods, cosmetics, and pharmaceuticals for a long time, owing to their unique physicochemical properties [11,12,13,14]. 

Naturally occurring polysaccharides rarely exhibit ideal characteristics consistent with their efficient use in industrial applications. As such, a range of approaches has been developed to alter the physicochemical and antioxidant properties of these polysaccharides in order to broaden their utility, including acid hydrolysis, alkaline hydrolysis, high-pressure homogenization, and ultrasound treatment [7,11,15,16,17]. Subcritical water (SCW) has been a focus of substantial research interest owing to its unique, temperature-controlled density, viscosity, dielectric constant, ionic product, diffusivity, electric conductance, and solvent properties [18]. SCW has recently been employed to extract bioactive compounds including polysaccharides, pectin, proteins, and polyphenols from raw food waste materials [19], including pumpkin by-products [20,21,22,23]. SCW was applied and optimized by some researchers in the extraction of polysaccharides with specific biological activities [24,25,26]. Lentinan and soybean protein structures can be altered by exposing them to SCW at increasingly high extraction temperatures [27,28], suggesting that such SCW treatment may represent a viable approach to biomacromolecule modification.

The antioxidant activity of polysaccharides is most commonly associated with their molecular weight, chain conformation, degree of modification, and physicochemical properties [29]. No prior studies have, as far as we are aware, evaluated the use of SCW treatment as an approach to polysaccharide modification. However, to expand the utility of such modified pumpkin polysaccharides, it is essential that their properties and antioxidant activity levels be analyzed following SCW treatment. 

## 2. Materials and Methods

### 2.1. Materials

Fresh mature pumpkin (*C. moschata*) fruits were purchased from a local market in Beijing, China. The shape, color, and weight of the selected pumpkin are basically the same.

### 2.2. Pumpkin Polysaccharide Preparation

Pumpkin polysaccharides were extracted with hot water using a modified version of a previously published protocol [30]. Briefly, pumpkins were cut into pieces, homogenized, and combined at a 1:4 ratio with distilled water. The resultant puree was incubated for 6 h at 85 °C, after which it was centrifuged. Then, the precipitate was combined with an equivalent volume of distilled water, and this heating procedure was repeated three times, after which the supernatants from these extraction steps were pooled, dialyzed against distilled water, and concentrated to one-quarter of the starting volume via evaporation at 40 °C. The concentrated solution was deproteinized using the trichloroacetic acid (TCA) method. Then, three volumes of 95% ethanol were added (final concentration: 80%) to the aqueous phase. Then, polysaccharide-containing precipitates were lyophilized at −20 °C for 48 h.

### 2.3. Subcritical Water Hydrolysis

All SCW hydrolysis was conducted in a 50-mL stainless steel reactor (Yanzheng, Shanghai, China). Briefly, 1 g of pumpkin polysaccharides was combined with 30 mL of distilled water. Then, the reactor was closed and heated to one of the tested SCW temperatures (120–210 °C in 30 °C increments) under a pressure of 10 MPa of pressure using a temperature controller and a pressure gauge, respectively. Nitrogen gas (99.99% pure) was used to maintain the pressure in the reactor, and samples were stirred with a magnetic stirrer and a high-temperature-resistant rotor. After samples were incubated for 10 min at the chosen SCW temperature, vessels were cooled with running tap water. Then, samples were collected from the reaction vessel, spun for 20 min at 10,000 g at 4 °C, and supernatants were precipitated as above via the addition of three volumes of 95% ethanol, after which the precipitates were lyophilized at −20 °C for 48 h.

### 2.4. Total Carbohydrate and Reducing Sugar Content Measurements

Total carbohydrates were measured using a slightly modified version of a previously described phenol–sulfuric acid method [31]. Briefly, 0.5 mL of hydrolysate samples were combined with 2.5 mL of concentrated sulfuric acid, after which 0.5 mL of (5%) phenol was added. Then, the reaction was incubated for 30 min at 100 °C in a water bath prior to cooling to room temperature. Then, absorbance was analyzed at 490 nm. Calibration curves were prepared using D-glucose. 

Reducing sugar content was measured via the 3, 5-dinitrosalicylic acid (DNS) method using a slightly modified version of a previous protocol [32]. Briefly, 1 g of NaOH was dissolved in 100 mL of distilled water, after which 1 g of DNS, 200 mg of phenol, and 50 mg of Na_2_SO_3_ were added to prepare the DNS reagent. Then, samples were analyzed by combining 3 mL of each sample with 3 mL of the DNS reagent prior to heating for 10 min in a 90 °C water bath to develop a reddish-brown color. Then, coloration was stabilized by adding 1 mL of a 40% potassium sodium tartrate (Rochelle salt) solution, and samples were cooled to room temperature. Then, absorbance at 575 nm for triplicate samples was measured, with D-glucose being used to prepare calibration curves.

### 2.5. Molecular Mass Determination

The molecular mass of the samples was determined following the method of Chen with some modification [15]. It was tested using a high-performance size exclusion chromatography (HPSEC) with a Shimadzu LC-10A HPLC system with a refractive index detector (RID) (Shimadzu, Japan). A column PL aquagel-OH MIXED (7.5 × 300 mm^2^) (Agilent, US) was used. Sample (20 μL) solution (5 mg/mL) filtered through a 0.22 μm membrane filter was injected in each run. The mobile phase was 0.1 M NaNO_3_ solution which contained 0.05% NaN_3_ as a preservative. The flow rate was 1 ml/min, and the temperature of the column was 40 °C.

### 2.6. Rheological Measurements

Samples of the isolated polysaccharides were dissolved in the deionized water at room temperature, after which the rheological properties of these solutions were assessed with a dynamic shear rheometer (Haake Mars 40, Thermo Scientific, Karlsruhe, Germany). For peak hold, steady-state flow, dynamic oscillation, and activation energy tests, polysaccharides were diluted to 0.1 g/mL. Before each measurement, the sample was allowed to equilibrate for 2 min.

#### 2.6.1. Intrinsic Viscosity and Viscosity-Average Molecular Weight

Polysaccharide intrinsic viscosity ([*η*]) was determined as described previously [33]. Briefly, viscosity values for a range of polysaccharide concentrations (0.08–0.16 g/mL) were assessed with a dynamic shear rheometer (Haake Mars 40, Thermo Scientific, Germany). A conical concentric cylinder with respective stator inner radius, rotor outer radius, and cylinder immersed height values of 15, 14, and 42 mm was used to conduct the peak hold test. The temperature during this testing was held constant at 25 °C using a water bath connected to a Peltier system, and the angular velocity was 100 rpm. The intrinsic viscosity ([*η*]) was estimated via linear extrapolation at a zero concentration using the following equations [34]:(1)$$ln\left(\frac{\eta_{sp}}{c}\right)=ln\left[\eta \right]+K\left[\eta \right]c$$
(2)$$\eta_{sp}=\frac{\eta -\eta_{0}}{\eta_{0}}$$
where [*η*] was the intrinsic viscosity (L/g), η_sp_ was the specific viscosity, c was the polysaccharide solution concentration (g/L), *η* was the viscosity of polysaccharides solutions (Pa s), *η_0_* was the solvent viscosity (deionized water) (Pa s), and K was Martin’s constant.

The viscosity-average molecular weight [*M_v_*] was determined with the Mark–Houwink–Sakurada equation:(3)$$\left[\eta \right]=kM_{v}^{\alpha }$$
where the constants *k* and *α* were 2.34 × 10^−5^ and 0.8224, respectively as per the solute–solvent system and temperature [35].

#### 2.6.2. Steady-State Flow Test

The shear rate during the steady-state flow step at 25 °C ranged from 0.1 to 100 s^−1^, which was conducted with an aluminum plate (diameter 40 mm, gap 1000 mm). Changes in apparent viscosity with shear rate were fitted to the Hershel–Bulkley models [36].
(4)$$\tau =\tau_{0}+k\cdot \gamma^{n}$$
where *τ* was the shear stress (Pa), *τ*_0_ was the yield stress (Pa), *γ* was the shear rate (s^−1^), and *k* and *n* were dimensionless flow behavior index and consistency coefficient values (Pa·s^n^), respectively.

#### 2.6.3. Dynamic Oscillation

The linear viscoelastic region (LVR) of this polysaccharide solution was assessed through strain sweep tests from 0.01 to 100% at 6.283 rad/s. For the frequency sweep step, the angular frequency ranged from 0.1 to 10 rad/s at 25 °C in the LVR (strain = 1%), and this test was conducted with an aluminum plate (diameter 40 mm, gap 1000 mm). Then, polysaccharide solution storage modulus (*G*’) values were approximated with the following formula: (5)$$G^{\prime }=K^{\prime }\cdot \omega^{n^{\prime }}$$
where *K’* was a constant, *n’* was the frequency, and *ω* was the angular frequency.

#### 2.6.4. Activation Energy

The activation energy was determined based upon changes in apparent viscosity with temperature as described previously [37]. A temperature ramp step from 10 to 50 °C was conducted at a constant angular velocity of 0.5 rad/s using a 40 mm diameter, 1000 mm gap aluminum plate, with the Arrhenius equation then being used to calculate activation energy: (6)$$\eta_{\alpha }=\eta_{\infty }exp\left(\frac{E_{\alpha }}{RT}\right)$$
where *η_α_* was the apparent viscosity (Pa.s) at an angular velocity of 0.5 rad/s, *η_∞_* was the dimensionless frequency factor constant, *E_α_* was the activation energy (kJ/mol), R was the ideal gas constant (8.3145 J/mol K), and T was the absolute temperature (K).

### 2.7. Emulsifying Properties

To prepare emulsions, 100 mL of polysaccharides (20 g/L) were combined with 5 g of corn oil with a density of 0.84 g/cm^3^. Then, pre-homogenization of these mixtures was conducted for 3 min with a high-speed emulsifier (Ultra-Turrax T 18; IKA, Staufen, Germany) at 12,000 rpm at 300 bar pressure in three passes through a homogenizer (FB-110Q, Litu, Shanghai, China).

#### 2.7.1. Emulsifying Activity

Emulsifying activity was determined using a slightly modified version of a previous protocol [38]. Briefly, emulsions were diluted to 900 dilutions using 1 g/L sodium dodecyl sulfate (SDS). Then, emulsion turbidity was assessed at 500 nm with a UV spectrophotometer (i9, Haineng, Jinan, China), with 1 g/L SDS serving as a blank. Turbidity (*T*) was calculated as follows:(7)$$T=\frac{2.303\cdot A\cdot V}{l}$$
where *V* was the dimensionless dilution factor, *A* was the absorbance at 500 nm, and I was the path length (0.01 m). 

The emulsion activity index (*EAI*) was determined as follows:(8)$$EAI=\frac{2T}{\emptyset \cdot c}$$
where *Ø* was the oil volume fraction of the dispersed phase, and *c* was the polysaccharide concentration in the emulsion.

#### 2.7.2. Zeta Potential and Conductivity

Polysaccharide emulsion zeta-potential values were assessed via dynamic light scattering with a Malvern Zetasizer Nano ZS instrument (ZEN3700, Malvern Instruments, Worcestershire, UK). To eliminate multiple scattering effects, emulsions were diluted to an oil droplet concentration of 0.053 g/L using pH-adjusted deionized water, after which they were injected into the zeta cell. Refractive indices for the oil droplets and solvent were 1.45 and 1.33, respectively. All testing was conducted at 25 °C with a 120 s equilibration time. Solution conductivity was measured simultaneously [39]. Emulsion droplet zeta-potential and conductivity were analyzed via electrophoresis and laser Doppler velocimetry based on the direction and rate of droplets within the applied electric field. 

### 2.8. Fourier Transform Infrared (FTIR) Spectroscopy and Scanning Electron Microscopy (SEM) Analyses

Samples of pumpkin polysaccharides (2 mg) were mixed with 100 mg of KBr and pressed into thin pellets, which were then subjected to FTIR with an appropriate spectrometer (Thermo Fisher Scientific) from 4000 to 400 cm^−1^.

Polysaccharide microstructural changes were assessed via SEM (Inspect F50, FEI, Hillsboro, OR, USA). Prior to analysis, samples were adhered to the sample stage and coated with a thin layer of gold, after which they were assessed at 0.5 k×, 1 k×, 2 k×, and 5 k×.

### 2.9. Antioxidant Activity Analyses

#### 2.9.1. Measurement of DPPH Radical Scavenging Activity

1,1-diphenyl-2-picrylhydrazyl (DPPH) radical scavenging analyses were conducted as detailed previously [40]. A 2 mL sample solution (6, 7, 8, 9, or 10 mg/mL) was mixed with 2 mL (0.1 mmol/L) of a DPPH ethanol solution for 30 min at 37 °C while protected from light. Then, absorbance at 517 nm was assessed, and DPPH radical scavenging rates were calculated as follows: (9)$$\mathrm{DPPH}\;\mathrm{radical}\;\mathrm{scavenging}\;\mathrm{rate}\;\left(\%\right)=1-\frac{A_{i}-A_{j}}{A_{c}}\times 100\%$$
where A_c_, A_i_, and A_j_ were the absorbance of the sample-free DPPH solution, a combination of the sample and the DPPH solution, and the DPPH-free sample, respectively. 

#### 2.9.2. Hydroxyl Radical Scavenging Activity Measurements

Polysaccharide hydroxyl radical scavenging activity was measured as in previous reports [41]. Briefly, test samples (100 μL at 1, 2, 3, 4, or 5 mg/mL) were combined with 50 μL of 1.5 mmol FeSO_4_, 35 μL of 6.7 mmol H_2_O_2_, and 15 μL of 20 mmol salicylic acid for 1 h at 37 °C while being protected from light. Then, absorbance at 562 nm was assessed, and ascorbic acid (Vc) was utilized as a positive control. Then, hydroxyl radical scavenging rates were calculated as follows: (10)$$\mathrm{Hydroxyl}\;\mathrm{radical}\;\mathrm{scavenging}\;\mathrm{rate}=\left\{1-\left(A_{1}-A_{2}\right)/A_{0}\}*100\right.\%$$
where *A*_0_, *A*_1_, and *A*_2_ were the absorbance of the sample-free reagent, the sample reaction solution, and a reagent blank solution free of salicylic acid, respectively.

#### 2.9.3. ABTS Radical Scavenging Assay

The ability of these polysaccharides to scavenge 2, 2’-azino-bis(3-ethylbenzothiazoline-6-sulfonic acid) (ABTS) radicals was determined as in prior studies [42]. Briefly, 7 mM of an ABTS solution was reacted for 16 h with 2.45 mM potassium persulfate at room temperature while protected from light, after which samples were diluted with phosphate buffer saline (PBS) (pH 7.0) to an absorbance of 0.700 ± 0.010. Then, a 0.15 mL volume of a sample solution (1, 2, 3, 4, or 5 mg/mL) was mixed with 2.85 mL of ABTS solution, and this mixture was incubated for 10 min at room temperature. Then, absorbance at 734 nm was assessed, and the ABTS radical scavenging rate was determined as follows: (11)$$\mathrm{ABTS}\;\mathrm{radical}\;\mathrm{scavenging}\;\mathrm{rate}\;\left(\%\right)=1-\frac{A_{s}-A_{x}}{A_{o}}\times 100\%$$
where A_o_, A_s_, and A_x_ were the absorbance of ABTS without sample, sample combined with ABTS, and sample without ABTS, respectively. 

#### 2.9.4. Reducing Power

Polysaccharide reducing power was determined as reported previously by Zhang et al. [43]. Briefly, a 1 mL volume of polysaccharide solution (1, 2, 3, 4, or 5 mg/mL) was combined with PBS (pH 6.6) and 2.5 mL 1% K_3_[Fe(CN)_6_] for 20 min at 50 °C. Then, the solution was cooled and combined with 2.5 mL of a 10% TCA solution. Then, this solution was spun for 10 min at 3000 rpm, and 2 mL of the supernatant was combined with 2 mL of deionized water and 1.0 mL of a 0.1% FeCl_3_ solution. Following a 10-min reaction at room temperature, the absorbance of this solution was assessed at 700 nm. 

### 2.10. Statistical Analysis

SAS 9.3 (SAS Institute, Cary, NC, USA) was used for all statistical testing. Data were means ± standard deviations (SD) (*n* = 3), and were compared via ANOVAs. *p* < 0.05 served as the significance threshold for these analyses. 

## 3. Results and Discussion

### 3.1. The Molecular Mass and Distribution

The number-average molecular weight (Mn) represents the general arithmetic average value of the molecular weight of the substance, and its change indicates the overall destruction of the polysaccharides chain. Weight-average molecular weight (Mw) emphasizes the contribution of bigger polysaccharides chains to Mw distribution as well as polymer physicochemical properties. The molecular weight of polysaccharides is dependent on the source and varies with different modification methods [44]. The physical and chemical properties of polysaccharides are different with different molecular weight [45]. Under the fixed temperatures and polysaccharide concentrations, molecular weight and distribution of Lentinan is the main factor that affects its gelling and thickening properties [46]. The molecular mass and distribution of pumpkin polysaccharides under different subcritical water conditions are shown in Figure 1A–E. SCW treatment decreased the Mw and Mn of polysaccharides (Mw changed from 3.21 × 10^5^ Da to 1.71 × 10^3^ Da, while Mn changed from 1.31 × 10^3^ Da to 7.07 × 10^2^ Da). The four peaks of pumpkin crude polysaccharides (PCP) showed that PCP was composed of fragments with different molecular weights. With the increase of treating temperature, peaks with high molecular weight diminished gradually, with only two peaks corresponding to low molecular weight component left in samples treated by SCW at 210 °C (Figure 1F). These results indicate that SCW is an efficient approach for degrading pumpkin polysaccharides.

The degradation of pumpkin polysaccharides was also evidenced by a decrease of total carbohydrate content and presence of reducing sugar in the SCW-treated samples. Small-molecule reducing sugars emerged during the hydrolysis of pumpkin polysaccharides, which might be decomposed or converted to other compounds subsequently [47,48]. This process is visually assessed by the color change, as displayed in Figure 1F. Pumpkin polysaccharide solution treated by SCW at 120 and 150 °C exhibited a light-yellow color, while those treated by SCW at 180 and 210 °C were more brownish.

### 3.2. Rheological Analyses

#### 3.2.1. Viscosity-Related Analyses

Intrinsic viscosity [*η*] corresponds to the parameters of given polymer mass that are occupied in the hydrodynamic volume, and it is a primary determinant of polymer drag reduction and extensional viscosity in a range of contexts [49]. Herein, we utilized the Martin equation and viscosity-average molecular weight (Mv) derived from the Mark–Houwink–Sakurada equation to describe intrinsic viscosity for our polysaccharide solutions (Figure 2C). The intrinsic viscosity [*η*] declined from 0.085 to 0.027 L/g with rising SCW temperature; in the meantime, the viscosity-average molecular weight (Mv) decreased from 21.25 to 5.37 kDa. Following higher temperature SCW treatment, these polysaccharides exhibited decreased intrinsic viscosity that may coincide with greater degradation and less molecular chain entanglement [12,50], which is likely due to the effective hydrolytic activity of SCW treatment. 

#### 3.2.2. Apparent Viscosity Analyses

Steady-state flow measurements were next used to assess the apparent viscosity of prepared pumpkin polysaccharide solutions (Figure 2A). We found that viscosity values for these solutions declined with rising shear rate across the 0.1 to 100 s^−1^, with all solutions exhibiting clear shear-thinning pseudoplastic behavior. SCW treatment greatly affected the viscosity and rheological properties of these polysaccharides’ solutions. Different from others, the apparent viscosity of SW150 decreases slowly with the increase of shear rate. The observed treatment-related drop in viscosity as a function of shear rate can be attributed to the reduced number of chain entanglements [51]. In addition, the different SCW treatment temperatures may have differentially impacted polysaccharide molecular weight values, in turn impacting the viscosity of solutions prepared using these polysaccharides [52]. Thus, subcritical hydrolysis-mediated changes in polysaccharide molecular weight may be the primary drivers of SCW-dependent changes in sample viscosity. 

The Herschel–Bulkley fluid model describes non-Newtonian fluids for which the relationship between stress and strain is non-linear and complex [53]. Herschel–Bulkley model parameters values for the present study are compiled in Table 1. High R^2^ values were generated when comparing predicted and experimental models, suggesting that this Herschel–Bulkley model can effectively describe the rheological behavior of SCW-treated pumpkin polysaccharide solutions. Yield stress (τ_0_) can be used to assess the minimum amount of force that must be applied to a fluid to initiate its movement [36], with differences in τ_0_ for the tested solutions corresponding to differences in apparent viscosity. The consistency coefficient (K) is a function of solution viscosity, and the flow behavior index (n) values for the tested solutions exhibited shear-thinning behavior consistent with the apparent viscosity curves shown in Figure 2A. 

#### 3.2.3. Strain Sweep Analysis

The linear viscoelastic region in which a system can stand without permanent structural damage was assessed through strain sweep analyses [54]. Storage modulus (G’) was plotted against strain ranged from 0.1 to 100% at 6.28 rad/s (Figure 2B). For PCP, SW120, and SW150, a similar plateau region of G’ value was observed prior to decreasing with increasing stress, while the plateau region for the SW180 sample was obviously shorter. In contrast, SW210 did not have a plateau region, and the G’ value declined with increasing stress from the beginning, indicating that the SW210 fluid was no longer in the linear viscoelastic region and that mechanical deformation was occurring. Our results suggested that high-temperature SCW treatment greatly impacts the linear viscoelastic properties of pumpkin polysaccharides solutions.

#### 3.2.4. Activation Energy

Activation energy (E_a_) can be analyzed to assess the sensitivity of apparent temperature-related viscosity changes [55]. From 10 to 50 °C, we found that apparent viscosity values for all polysaccharides declined with rising temperature, indicating that higher temperatures improved flow performance. Related E_a_ values for these solutions are shown in Figure 2D. As the SCW treatment temperature increased, the E_a_ value for these pumpkin polysaccharides rose from 22.59 to 63.61 kJ/mol, with values coinciding with apparent viscosity within the analyzed temperature range. The SCW temperature impacted polysaccharide activation energy, which corresponds to interactions between polysaccharide chains [12]. Specifically, we found that SW210 samples exhibited maximal activation energy values and increased temperature sensitivity, which is consistent with a high degree of interaction between polymer chains [56].

### 3.3. Emulsifying Properties

The emulsifying activity index (EAI), zeta potential, and emulsion conductivity were measured to assess if SCW treatment can improve the property of pumpkin polysaccharides as an emulsifier. The SW150 sample possessed the highest EAI value (53.0 m^2^/g), followed by SW120 and SW180, which were both significantly higher than the untreated sample (PCP). This result indicated that SCW treatment can significantly increase the emulsifying activity of pumpkin polysaccharides. Zeta potential is an analytical technique that is applied for the determination of surface charge of nanoparticles in colloidal solution. The magnitude of zeta potential gives a prediction of the colloidal stability. Zeta potential of nanoparticles with absolute values greater than 25 mV have a high degree of stability [57]. Pumpkin polysaccharides were negatively charged on the interface, with zeta potential values ranging from −26.5 to −33.0 mV (Figure 3B), suggesting good emulsifying properties [58]. Emulsion conductivity was measured to provide insight into the stability of these emulsions [59]. Similar conductivity values were observed for all emulsions (Figure 3C), ranging from 0.014 to 0.022 mS/cm, which indicated that all of these emulsions exhibited good stability. Taken together, our results indicate that SCW-treated pumpkin polysaccharides exhibited an enhanced emulsifying activity and acceptable emulsion stability.

### 3.4. FTIR Analyses of SCW-Treated Polysaccharides

FTIR spectra of the analyzed pumpkin polysaccharides are shown in Figure 4A. A broad peak evident at 3299 cm^−1^ represents a characteristic O-H stretching vibration absorption peak, and as polysaccharides contain many hydroxyl groups that can mediate intra- and intermolecular binding, this peak was notably broad. The absorption peak at 2925 cm^−1^ is correlated to the C-H stretching vibrations, whereas the peaks at 1606 cm^−1^ and 1411 cm^−1^ represent carboxyl group asymmetric and symmetric stretching vibrations, respectively, confirming that uronic acid is present in pumpkin polysaccharide. The absorption peak at 1000–1200 cm^−1^ was attributable to the vibration of C-O and C-C in the sugar ring backbone, while peaks at 896 and 833 cm^−1^ suggested that these pumpkin polysaccharides exhibited both β and α configurations [10,60].

### 3.5. Scanning Electron Microscopy

Next, SEM was used to assess microstructural changes in these polysaccharides following SCW treatment (Figure 4B). Pumpkin polysaccharides without SCW treatment showed irregular sheet-like structures (Figure 4B(A1–D4)). With SCW exposure, a clear fragmentation of polysaccharide particles indicated the degradation of pumpkin polysaccharide. This is in line with the observed treatment-related reductions in molecular weight and suggests that SCW can efficiently modify the microstructure of pumpkin polysaccharides. The sample under SCW 150 °C treatment exhibited the finest particle size, indicating larger surface areas (Figure 4B(C1–C4)). SCW treatment at higher or lower temperature either had a lower degree of fragmentation or caused some kind of aggregation, respectively. The aggregation at high temperature could be related to the smoother particle surface and more inter-molecule hydrogen bonds formed among low molecular weight sugars in SCW180 and SCW210 (Figure 4B(B1–E4,D1–D4,E1–E4)) [61,62].

### 3.6. Assessment of Pumpkin Polysaccharide Antioxidant Activity

Lastly, we assessed the antioxidant activity of these SCW-treated pumpkin polysaccharide preparations in vitro through hydroxyl radical scavenging, ABTS free radical scavenging, DPPH free radical scavenging, and reducing power assays. A concentration dependent increase of antioxidant activity was observed for all samples (Figure 5). All SCW-treated samples exhibited enhanced hydroxyl radical scavenging rates (Figure 5A), with IC_50_ values for pumpkin crude polysaccharides (PCP) and the SW120, SW150, SW180, SW210 of 6.22, 1.68, 0.08, 0.26, and 0.39 mg/mL, respectively. Compared with the untreated sample, SW150 and SW180 samples had higher ABTS radical scavenging rates (Figure 5B), with a respective 54.57% and 25.7% scavenging rate at a concentration of 1 mg/mL, respectively. SCW treatment also improved the DPPH clearance rates of pumpkin polysaccharide (Figure 5C) as follows: SW150 (IC_50_ = 6.32 mg/mL) > SW180 (IC_50_ = 7.06 mg/mL) > SW120 (IC_50_ = 9.05 mg/mL) > SW210 (IC_50_ = 10.26 mg/mL) > PCP (IC_50_ = 20.85 mg/mL). Similar results were obtained when measuring total reducing power (Figure 5D). Overall, pumpkin polysaccharides treated by SCW at 150 °C exhibited the highest antioxidant activity. This is likely due to a broader molecular weight distribution and larger surface area of SW150 samples as observed previously. Fragmentation by SCW can increase the number of active reducing end of pumpkin polysaccharides, thus increasing the antioxidant capacity. Our results also suggest that the molecular weight is a key determinant for the antioxidant activity of polysaccharides, and SCW treatment is an effective approach of modifying pumpkin polysaccharides to achieve better bioactivity [1,63,64].

## 4. Conclusions

In summary, we found that SCW treatments across a range of temperatures (from 120–210 °C) can modify the rheological, structural, and emulsifying properties of pumpkin polysaccharides. Pumpkin polysaccharides solution exhibited pseudo-plastic flow-like behavior, and SCW treatments altered the apparent viscosity and activation energy of these samples. Specifically, polysaccharides treated with SCW at 150 °C exhibited good emulsifying activity and enhanced antioxidant activity compared with native pumpkin polysaccharides, which was likely due to a broader molecular mass distribution and more reducing ends generated by SCW. SEM analyses suggested that SCW treatment altered the morphology of these polysaccharides, resulting in a bigger surface area being exposed. Taken together, our research indicated that SCW treatments represent a simple, tunable approach to partially degrading pumpkin polysaccharides and thereby modifying their physicochemical and bioactive properties.

## Figures and Tables

**Figure 1 foods-10-00197-f001:**
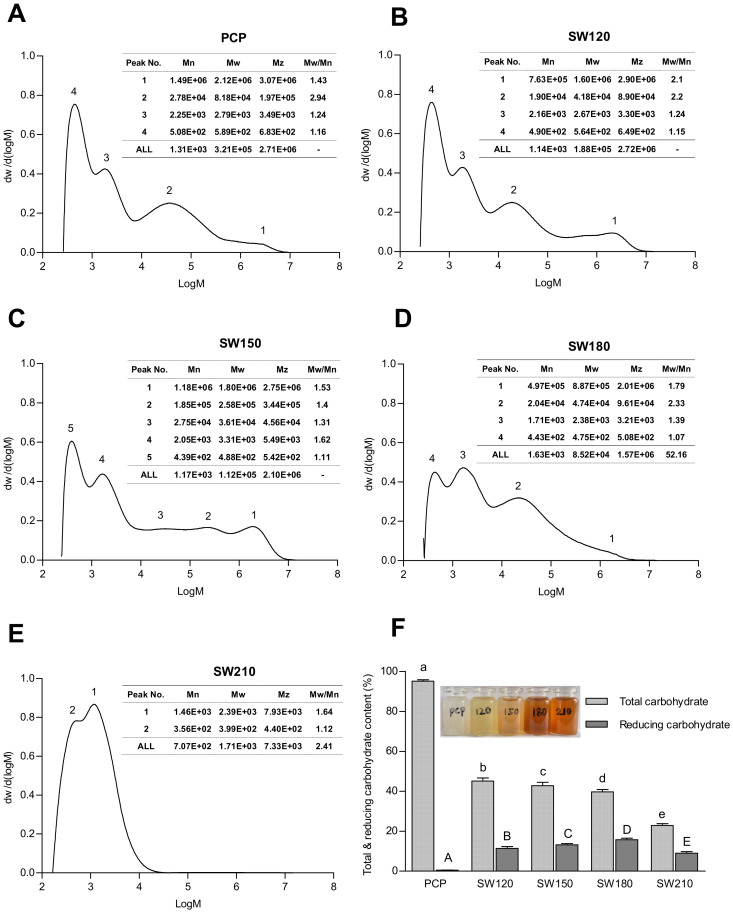
The molecular mass distribution of PCP (**A**), SW120 (**B**), SW150 (**C**), SW180 (**D**), SW210 (**E**), and total carbohydrate and reducing carbohydrate (**F**) of polysaccharides under different subcritical water conditions. Data were expressed by means ± standard deviations (*n* = 3). Values with different letters are significantly different (*p* < 0.05). PCP, pumpkin crude polysaccharides; SW120, PCP under subcritical water 120 °C treatment; SW150, PCP under subcritical water 150 °C treatment; SW180, PCP under subcritical water 180 °C treatment; SW210, PCP under subcritical water 210 °C treatment.

**Figure 2 foods-10-00197-f002:**
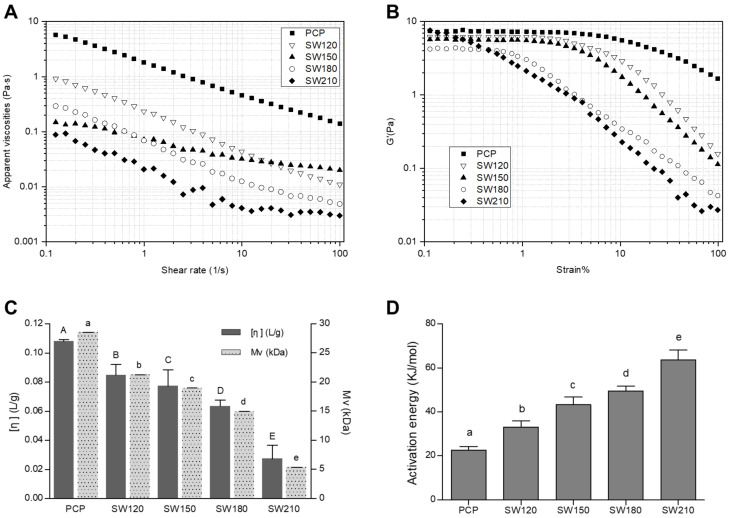
Apparent viscosities (**A**), storage modulus (G’) curves (**B**), intrinsic viscosity ([*η*]) and viscosity-average molecular weight (Mv) (**C**), and activation energy (**D**) of variety polysaccharides solutions under different SCW conditions. Data were expressed by means ± standard deviations (*n* = 3). Values with different letters are significantly different (*p* < 0.05). PCP, pumpkin crude polysaccharides; SW120, PCP under subcritical water 120 °C treatment; SW150, PCP under subcritical water 150 °C treatment; SW180, PCP under subcritical water 180 °C treatment; SW210, PCP under subcritical water 210 °C treatment.

**Figure 3 foods-10-00197-f003:**
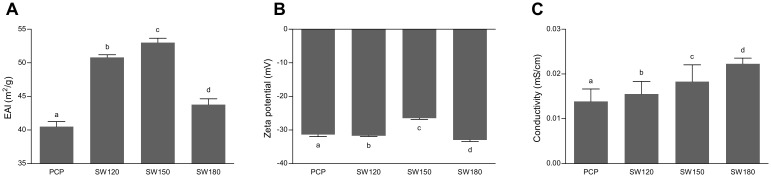
Emulsifying activity (**A**), Zeta potential (**B**), and Conductivity (**C**) of polysaccharides emulsions under different SCW conditions. Data were expressed by means ± SD (*n* = 3). Values in the same column with different letters are significantly different (*p* < 0.05). PCP, pumpkin crude polysaccharides; SW120, PCP under SCW 120 °C treatment; SW150, PCP under SCW 150 °C treatment; SW180, PCP under SCW 180 °C treatment.

**Figure 4 foods-10-00197-f004:**
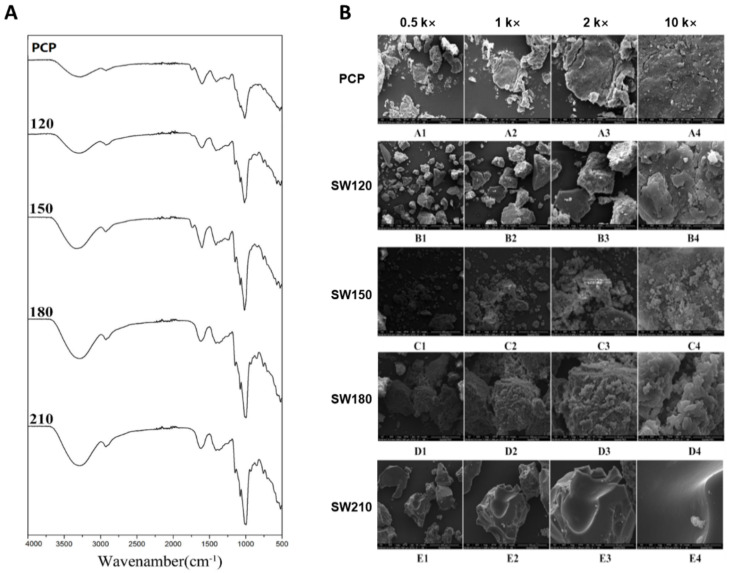
FTIR spectra (**A**) and scanning electron microphotograph (SEM) (**B**) of pumpkin polysaccharides treated with different subcritical water (SCW) conditions.

**Figure 5 foods-10-00197-f005:**
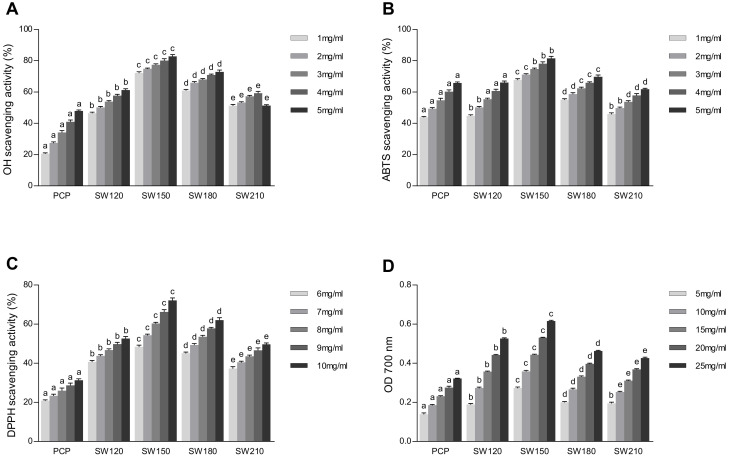
Scavenging effect of pumpkin polysaccharides on hydroxyl radical (**A**), ABTS radical (**B**), DPPH radical (**C**), and reducing power (**D**). Data were expressed by means ± SD (*n* = 3). Values with different letters in the same concentration column differ significantly (*p* < 0.05). PCP, pumpkin crude polysaccharides; SW120, PCP under SCW 120 °C treatment; SW150, PCP under SCW 150 °C treatment; SW180, PCP under SCW 180 °C treatment.

**Table 1 foods-10-00197-t001:** Herschel–Bulkley model parameters of polysaccharides solutions (0.1 g/mL) under different SCW conditions.

Sample	τ_0_(Pa)	K (Pa·s^n^)	n	R^2^
PCP	0.440 ± 0.039 ^a^	1.309 ± 0.032 ^a^	0.206 ± 0.005 ^a^	1.000
SW120	0.153 ± 0.003 ^b^	0.047 ± 0.001 ^b^	0.416 ± 0.006 ^b^	1.000
SW150	0.121 ± 0.014 ^c^	0.023 ± 0.012 ^c^	0.480 ± 0.025 ^c^	0.991
SW180	0.050 ± 0.003 ^d^	0.013 ± 0.001 ^d^	0.758 ± 0.023 ^d^	0.995
SW210	0.015 ± 0.002 ^e^	0.003 ± 0.001 ^e^	0.964 ± 0.033 ^e^	0.993

Data were expressed by means ± SD (*n* = 3). Values in the same column with different letters are significantly different (*p* < 0.05). PCP, pumpkin crude polysaccharides; SW120, PCP under SCW 120 °C treatment; SW150, PCP under SCW 150 °C treatment; SW180, PCP under SCW 180 °C treatment; SW210, PCP under SCW 210 °C treatment; SW240, PCP under SCW 240 °C treatment.

## Data Availability

Not applicable.

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
