# Peer review of "Physicochemical Properties and Antioxidant Activity of Pumpkin Polysaccharide (Cucurbita moschata Duchesne ex Poiret) Modified by Subcritical Water"

_foods, 2021, doi:10.3390/foods10010197_

Round 1

Reviewer 1 Report

Authors describe the extraction and modification of pumpkin polysaccharides by hot water and subcritical water. The properties of the modified polysaccharides are studies and characterized. With the amount of hydrolysis, the molecular weight is decreasing in parallel with the viscosity. Also there is a decrease in antioxidant activit that is a pity. Overall, the novelty and the significance of these results are questionable. 

Appl. Sci. 2020, 10, 8815; doi:10.3390/app10248815

J Food Sci Technol (2020). https://doi.org/10.1007/s13197-020-04624-x

https://doi.org/10.1016/j.ijbiomac.2018.06.148

Prev. Nutr. Food Sci. 2016;21(2):132-137

10.1007/s11694-020-00491-4

Authors are suggested to include the following papers to the intorduction, and if already included than answer the corresponding questions below.

  • it is reported that these type of extraction and manipulation methods are useful for the isolation of many type of compounds (e.g. phenolic, protein etc) not only polysaccharides. How can the authors be sure that in their extract only polysaccharides are showing the antioxidant effect?
  • Why is it important and what is the significance of the SCW treatment, if the advantageous effect is lost?
  • Zeta potential of SW150 is much different than that of all the others, although in other properties there is no such difference. How can the authors address this issue?

Reviewer 2 Report

Manuscript ID: foods-1073615

Title: Characterization and antioxidant activity of pumpkin polysaccharide (Cucurbita moschata) modified by subcritical water

Authors: Guoyong Yu , Jing Zhao , Yunlu Wei , Linlin Huang , Fei Li , Yu Zhang , Quanhong Li

Overview and general recommendation:

In the manuscript the characterization of physicochemical properties and antioxidant activity of pumpkin polysaccharide modified by subcritical water (SCW). Although the characteristic and antioxidant properties od pumpkin polysaccharides are well known and described in the literature, the modification such as subcritical extraction is a innovation.

The researches are well organized, well described and supported with the literature discussion.

Below I give some concerns that require review:

Minor comments

  1. Title –.”Characterization and antioxidant activity of pumpkin polysaccharide (Cucurbita moschata) modified by subcritical water” – why authors have decided to separate antioxidant properties – isn’t it a part of characterization? If authors wish to emphasize this particular properties it might be convenient to change also the “characterisation” into specific properties by giving detail name?
  2. Literature sources that are cited in the text – at this stage of preparing the manuscript it is not that important, but please change the references format according to the Foods journal guidelines.
  3. “Historically, polysaccharides have also been leveraged as stabilizing, thickening, binding, and emulsifying agents in the industrial production of foods, cosmetics, and pharmaceuticals owing to their unique physicochemical properties (Liang et al., 2018) (Maalej, Hmidet, Boisset, Bayma, & Nasri, 2016) (Ji et al., 2019) (Kumar & Mody, 2009)”are does properties not in use anymore? Why is sentence started with the historically?
  4. As SCW pumpkin polysaccharides extraction is not described in a literature it would be worth to give a characterisation of this process in the Introduction part.
  5. 2 Pumpkin polysaccharide preparation and 2.3. Subcritical water hydrolysishow was concentration via evaporation done? and what were the lyophilization process parameters?
  6. 5. Molecular mass determination – please add the name and country of the producer of chromatograph
  7. 6. Rheological measurements – please add the parameters of the measurement
  8. Polysaccharides with proper Mw has been reported to have better thickening and gelling properties (Hua, Yang, Din, Chi, & Yang, 2018).” – what proper refers to? Can it be point or compared with something?
  9. 1 – as during the writing of the results and discussion part authors compare picks, in my opinion graphs scale should on all A-E charts be the same to enable comparison. It also would be interesting if authors give a photo also of a PCP, what was the colour?
  10. 5. Scanning electron microscopy – the changing of the structure is very interesting, can authors give an explanation what might be responsible for this aggregation, smother structure?
  11. 6. Assessment of pumpkin polysaccharide antioxidant activity, figure 5 – there is a lack of statistical analysis – please add
  12. Conclusion Conclusions

Reviewer 3 Report

Dear Authors,

           Please find an enclosed review of the manuscript (Ms. Ref. No. foods-1073615) entitled Characterization and antioxidant activity of pumpkin polysaccharide (Cucurbita moschata) modified by subcritical water . Thank you for having me as a referee for the above-mentioned manuscript.

After studying the article given to me for evaluation, I state the following:

In the beginning, I would like to express words of my appreciation for the idea and effort put into conducting research and writing the manuscript recommended to me for review. The authors present a subcritical water application to modify pumpkin polysaccharides for investigating the properties and antioxidant activity in this plant. The topic introduces novelty and is well prepared.

However, after reading the manuscript, I have concerns regarding the issues listed below. I would like to ask you to elucidate the vagueness that emerged while reviewing the manuscript so you would be able to improve your investigation.

  1. In section 2.1, the authors describe pumpkins using parameters such as color, weight, and shape. There is no discussion of the ripeness of these fruits in the manuscript. In supermarkets, there are not only fruits to consume. The ripeness of each fruit is important due to the content of polysaccharides and other compounds. Therefore, please complete the weight of these fruits so that the reader can somehow refer to the results presented in the manuscript. In addition, I would ask you to include a representative picture of the fruit taken for the investigation. It will easily illustrate the state of ripeness of the fruit.
  2. In Section 2.2, please avoid repeating the same forms when using the words. In this case “using”, try to find another alternative word.
  3. In Section 2.3, the supercritical water pressure parameter was exactly 10 MPa or about ~10 MPa?
  4. In Section 2.5, the mobile phase was 40 oC, or the chromatographic column was thermostated at 40 oC? Please add a literature reference to the 0.1M NaNO3 mobile phase used to make it clear that this is a non-destructive solution for analysis.
  5. In Section 3.2.3, please do not use a shortened version of objections “didn’t”. Please use the form “did not”.
  6. Please use the proper format of citation reserved for the Foods journal.

Once again, congratulations on an interesting idea and the delivery of new results that will give a better understanding of extraction with water in the supercritical state.

Good luck!

Reviewer 4 Report

The paper “Characterization and antioxidant activity of pumpkin polysaccharide (Cucurbita moschata) modified by subcritical water” is interesting.

Few points should be improve:

  • The first time that cite a botanical name of a plant, it is necessary including its botanical author. Please, add this information for all cited plants. Only the first time the botanical name must be written extended (species and genus), subsequently the name of the genus must be written with a bullet. For example C. moschata without botanical name.
  • The data are shown with different decimal places. Standardize all data with the same decimal places.
  • Reduce self-citations.
  • There are some English corrections, typo errors, space omitted or double between word and word, points omitted at the end of sentences, double words, and incorrect parentheses.

Round 2

Reviewer 1 Report

Authors have addressed all the comments

This manuscript is a resubmission of an earlier submission. The following is a list of the peer review reports and author responses from that submission.